# Fabricating Planar Perovskite Solar Cells through a Greener Approach

**DOI:** 10.3390/nano14070594

**Published:** 2024-03-28

**Authors:** Sajid Sajid, Salem Alzahmi, Nouar Tabet, Yousef Haik, Ihab M. Obaidat

**Affiliations:** 1Department of Chemical & Petroleum Engineering, United Arab Emirates University, Al Ain P.O. Box 15551, United Arab Emirates; fary_sajjo@yahoo.com; 2National Water and Energy Center, United Arab Emirates University, Al Ain P.O. Box 15551, United Arab Emirates; 3Department of Applied Physics and Astronomy, University of Sharjah, Sharjah P.O. Box 27272, United Arab Emirates; ntabet@sharjah.ac.ae; 4Department of Mechanical and Nuclear Engineering, University of Sharjah, Sharjah P.O. Box 27272, United Arab Emirates; yhaik@sharjah.ac.ae; 5Department of Mechanical Engineering, The University of Jordan, Amman P.O. Box 11942, Jordan

**Keywords:** green solvent, triethyl phosphate, perovskite thin film, efficient perovskite solar cell

## Abstract

High-quality perovskite thin films are typically produced via solvent engineering, which results in efficient perovskite solar cells (PSCs). Nevertheless, the use of hazardous solvents like precursor solvents (N-Methyl-2-pyrrolidone (NMP), dimethyl sulfoxide (DMSO), dimethylformamide (DMF), gamma-butyrolactone (GBL)) and antisolvents (chlorobenzene (CB), dibutyl ether (DEE), diethyl ether (Et_2_O), etc.) is crucial to the preparation of perovskite solutions and the control of perovskite thin film crystallization. The consumption of hazardous solvents poses an imminent threat to both the health of manufacturers and the environment. Consequently, before PSCs are commercialized, the current concerns about the toxicity of solvents must be addressed. In this study, we fabricated highly efficient planar PSCs using a novel, environmentally friendly method. Initially, we employed a greener solvent engineering approach that substituted the hazardous precursor solvents with an environmentally friendly solvent called triethyl phosphate (TEP). In the following stage, we fabricated perovskite thin films without the use of an antisolvent by employing a two-step procedure. Of all the greener techniques used to fabricate PSCs, the FTO/SnO_2_/MAFAPbI_3_/spiro-OMeTAD planar device configuration yielded the highest PCE of 20.98%. Therefore, this work addresses the toxicity of the solvents used in the perovskite film fabrication procedure and provides a promising universal method for producing PSCs with high efficiency. The aforementioned environmentally friendly approach might allow for PSC fabrication on an industrial scale in the future under sustainable conditions.

## 1. Introduction

Organic–inorganic halide perovskites have been extensively used in light-emitting diodes, lasers, photovoltaics, and other fields in the past few decades [1,2,3,4]. Perovskite solar cells (PSCs) have seen an increase in power conversion efficiency (PCE) from 3.8% to 26% in the past several years, particularly in the photovoltaic area [5]. The desirable characteristics of perovskites for solar energy conversion, such as their balanced electron-and hole-transport lengths [6], high optical absorption coefficient [7], low exciton binding energies [8], and small carrier effective masses [9], can be attributed to some of the factors driving PSCs’ rapid progress. As Pb-based PSCs progress toward greater stability and high efficiency, they are exhibiting considerable promise for use in future applications. However, the inherent toxicity of Pb-based perovskites poses a serious threat to the environment and human health [10]. Pb poisoning of soil and water supplies has been shown to have a very serious negative impact on plant, animal, and human health [11]. Most people show signs of Pb poisoning when they consume up to 0.5 mg of lead per day [12].

Along with the unique properties of perovskites, rapid advancements in the process of obtaining superior perovskite films can be credited for the increase in PSC performance [13,14]. Out of all the manufacturing strategies for perovskite films, solvent engineering [15], which was first presented by Jeon et al. in 2014, has become one of the most important methods for fabricating exceptional perovskite films. Subsequently, many attempts have been made to improve the efficiency of PSCs, most notably the development of novel solvent systems and perovskite precursors [16]. Dimethyl sulfoxide (DMSO), gamma-butyrolactone (GBL), N-Methyl-2-pyrrolidone (NMP), and N,N-dimethyformamide (DMF) are commonly used in solvent engineering to prepare perovskite precursor solutions. DMSO serves as a ligand solvent due to its efficient coordination ability, while DMF or NMP or GBL act as host solvents due to their excellent dissolving capabilities [17]. In the meantime, antisolvents such as diethyl ether (Et_2_O), dibutyl ether (DEE), toluene, and chlorobenzene (CB) are frequently selected due to their miscible nature with DMF, NMP, DMSO, and GBL and their high insoluble features with perovskite [18,19]. Researchers have evaluated the possible effects of several solvents utilized for the fabrication of perovskite films on populations using a measure of disability-adjusted life year (DALY) that gauges health issues [20,21,22]. The greater the solvent’s DALY value, the higher the risks to human health. According to their findings, DMF has the highest DALY value, followed by NMP, GBL, and DMSO [22]. This suggests that the solvents frequently employed to prepare perovskite solutions, particularly DMF, could potentially have detrimental impacts on people’s health. Furthermore, by rapidly eliminating the host solvent, antisolvents are frequently utilized to promote the nucleation and crystallization of perovskite films [23,24]. However, the antisolvents that are widely utilized are extremely volatile and hazardous. As an example, CB and toluene are hazardous chemicals that are harmful to aquatic life and can have detrimental impacts on the underwater environment. According to a number of published papers, the toxicity of CB and toluene is a serious concern, as is the risk they pose to aquatic organisms [25,26]. Even more seriously, exposure to CB and toluene can cause a variety of adverse health consequences in people, such as headaches, nausea, fatigue, and discomfort to the eyes and upper respiratory system [27,28]. Prior studies have primarily focused on the removal of hazardous host solvents, and some have even demonstrated how to fabricate planar PSCs without using a harmful antisolvent [29]. A number of reports have demonstrated that the use of precursor solvents and antisolvents can also be eliminated by employing chemical or physical vapor deposition. Even so, there are a few drawbacks to the thermal evaporation process that prevent its wider use, including its low material utilization, high equipment costs, and high energy consumption [30,31]. Nevertheless, no study has simultaneously eliminated hazardous precursor solvents (e.g., DMF and DMSO) and antisolvents (e.g., CB, toluene, Et2O, and DEE). In order to ensure the protection of both the environment and the health of the workers involved in manufacturing, it is thus highly desirable to implement a green approach that can eliminate both toxic precursor solvents and antisolvents and, subsequently, manufacture high-quality perovskite thin films and produce highly efficient PSCs.

In this work, we present a novel greener process for concurrently removing solvent toxicity from both the precursor solvent and antisolvent during the formation of perovskite thin films and fabrication of efficient PSCs. In the first step of this innovative process, a non-toxic triethyl phosphate (TEP) solvent was used to prepare the PbI_2_ layer. To avoid using an antisolvent in the second step, FAMAI was dissolved in the TEP on top of the PbI_2_ layer. The planar PSC standard configuration FTO/SnO_2_/MAFAPbI_3_/Spiro-OMetAD/Au showed an excellent PCE of 20.98%, which is the highest value of the PSCs obtained through the green method for perovskite thin film preparation. As a result, this work not only offers a novel method of making perovskite thin films but also tackles the toxicity of the solvents used in the common fabrication process, paving the way for the environmentally friendly production of PSCs in the near future.

## 2. Materials and Methods

### 2.1. Preparation of the Precursor Solutions

The precursor solution for SnO_2_ was prepared by diluting 1 mL of SnO_2_ colloidal solution with 5 mL of deionized water. To prepare the PbI2 precursor, 691.5 mg of PbI_2_ [29] was dissolved in 900 µL of NMP, TEP, and DMF, respectively, and the mixtures were stirred for 2 h at 70 °C. Then, 1 mL of TEP was used to dissolve 90 mg of formamidinium iodide (FAI) and 6.4 mg of methylammonium iodide (MAI) for the MAFAI precursor solution. The solution for the hole-transporting material was made up of 80 mg of Spiro-OMeTAD in 1 mL of chlorobenzene, 28.5 μL of tert-butylpyridine in 1 mL of chlorobenzene, and 8.75 mg/mL of lithium bis-(trifluoromethanesulfonyl)-imide. In a glove box, this mixture was stirred for 6 h.

### 2.2. Fabrication of the Perovskite Solar Cells

The FTO substrates were cleaned sequentially by sonicating them in detergent solution, deionized water, ethanol, and isopropyl alcohol for 20 min each. After being nitrogen blow dried, the FTO substrates were treated with UV ozone for 15 min. The previously prepared SnO_2_ solution was dropped and spun for 15 s at 3000 rpm on the completely cleaned FTO substrate. The SnO_2_-coated FTO substrates were heated to 150 °C for 30 min. The FTO/SnO_2_ substrates were then placed in a glove box once they had cooled to room temperature. The PbI_2_ solution was spin-coated on an FTO/SnO_2_ substrate for 30 s at 1500 rpm, annealed for 60 s at 70 °C, and then cooled to room temperature. The prepared MAFAI precursor solution was spin-coated onto PbI_2_ for 30 s at 1800 rpm. Then, the perovskite precursor film was removed from the glove box and exposed to room temperature for 15 min at 150 °C while maintaining a humidity of 30–40%. It is important to take into account that considerable amounts of humidity are necessary to produce high-quality perovskite films and high-performance devices [29]. This could be explained by the possibility that the perovskite precursor will draw water to the surface, encouraging the growth of perovskite crystals [32]. The samples were moved to a glove box for additional processing following the formation of perovskites. After cooling to room temperature, the as-prepared perovskite films were spin-coated with Spiro-OMeTAD solution for 30 s at 4000 rpm. The solar cells were then finished by evaporating 80 nm gold electrodes onto the as-prepared thin films. For a 0.1 cm^2^ active area, a metal mask was employed in each device.

### 2.3. Device Characterization

Utilizing a scanning electron microscope (Hitachi S-4800), morphologies were investigated. The perovskite’s crystallinity was examined using an X-ray diffractometer (Bruker D8 Advance, MA, USA, Cu-Kα radiation of 0.15406 nm). A UV-Vis spectrophotometer (UV-2600) was used to determine the absorption spectrum. The obtained perovskite films’ steady PL spectra were analyzed using Edinburg PLS 980. With a 485 nm laser, a transient-state spectrophotometer (Edinburg Ins. F900) was used to measure the TRPL decay of the perovskite films. The J-V characteristic curves were obtained under AM 1.5G illumination with a power intensity of 100 mW cm^−2^ using a source meter (Keithley 2400) and forward (−0.1 to 1.2 V) scans from a solar simulator (XES-301S+EL-100). The step voltage was set to 12 mV, and the delay duration was set to 10 ms. A ZAHNER Zennium electrochemical workstation was used to measure the EIS. The LiCan9 Plus (LABOAO) FTIR spectrometer was used to gather the FT-IR spectra.

## 3. Results and Discussions

Figure 1 describes the novel greener approach used in this study, the details of which are given in the Materials and Methods section. As discussed before, most of the high-quality perovskite layer and efficient PSCs are usually fabricated using toxic precursor solvents and antisolvents such as NMP, DMF, CB, and DEE [29,33]. Therefore, it is vital to remove hazardous precursor solvents and toxic antisolvents. In this scenario, we replaced the toxic precursor solvents with a green TEP solvent and additionally utilized a two-step process to eliminate the toxic antisolvent. As shown in Table 1, the TEP has less harmful effects on safety, health, and the environment than the commonly used toxic solvents in the perovskite precursors. According to previous reports [34,35], the donor number (DN) of a solvent is an effective parameter for measuring the ability of solvents to solubilize perovskites. In this context, NMP, DMF, and TEP possess nearly equal DNs, with 27.3, 26.6 and 26.0, respectively [34]. Therefore, the almost equal DN of TEP indicates that it has a comparable dissolving capability to NMP and DMF for perovskite precursors, making it a green solvent able to replace toxic precursor solvents. The perovskite thin films prepared from dissolving PbI_2_ in NMP or TEP or DMF are shown in Figure 1. After spin-coating the precursor solution of FAI/MAI in TEP on the surface of the PbI_2_ films (prepared using NMP or TEP or DMF), the thin films were annealed at 150 °C for 15 min. The perovskite films show a black color after annealing, as depicted in the photographs of Figure 1.

Before using the as-stated approach for the fabrication of PSCs, it is important to investigate the morphology of the perovskite thin films. Therefore, scanning electron microscopy (SEM) was used to study the morphology of the perovskite thin films obtained with the two-step method without using any antisolvents. Figure 2a–c show the PbI_2_ thin films obtained from the precursor solutions containing NMP, TEP, and DMF, respectively. From the SEM images in Figure 2a–c, it is clear that the PbI_2_ films have enough space for the infiltration of the FAI/MAI precursor solution (prepared in the TEP solvent). The SEM images taken after the annealing of the NMP-, TEP-, and DMF-based perovskite intermediate phases show not only full surface coverage and homogeneity but also large grains with fewer boundaries, as can be seen in Figure 2d–f. The large grains with fewer boundaries and pinhole-free surface of the as-prepared perovskite layers are beneficial for minimizing the trap sites for the photogenerated charge carriers, as reported in numerous reports [36,37,38]. This indicates that the green solvent TEP is as effective as NMP and DMF in preparing a high-quality perovskite layer after annealing.

The possible interaction between the solvent of TEP and PbI_2_ could be a Lewis acid–base reaction, where TEP acts as a Lewis base, and Pb^2+^ acts as a Lewis acid. The FTIR transmittance spectra were used to investigate this interaction between the TEP and perovskite precursors. As depicted in Figure 3, there is a characteristic peak at 1322.4 cm^−1^, which corresponds to the stretching vibration of the P=O bond in TEP (see the blue line in Figure 3) [39]. The similar characteristic peak of the P=O bond also appears in the intermediate phase of the MAFAPbI_3_ films (see red line in Figure 3), indicating a solvent complex for the intermediate-phase films (here, FAI-MAI-PbI_2_-TEP). The solvent complex then transforms to perovskite films (FAMAPbI_3_) when TEP is evaporated through thermal treatment, which is evidenced by the absence of the characteristic peak of the P=O bond in the resultant perovskite films (see the black line in Figure 3). It was found that the stretching vibration peak of the P=O bond in the intermediate phase downshifted from 1322.4 cm^−1^ to 1279.3 cm^−1^ compared to that of TEP. The decreased stretching vibration peak of the P=O bond shows a weak strength between phosphorous and oxygen [40,41], which is due to the formation of a dative Pb=O bond between the TEP and Pb^2+^ via a Lewis acid–base reaction, thereby forming a Lewis acid–base adduct.

X-ray diffraction (XRD) measurements were carried out to assess the crystal structure of the perovskite thin films, as shown in Figure 4a. The XRD patterns of the as-prepared perovskite thin films can be correlated with the pure phase of the material with the (110) and (220) planes at 14^0^ and 28^0^. The fact that all the perovskite thin films have excellent crystals with almost identical intensity peaks is significant because it indicates large grains with few boundaries. This is in accordance with prior studies that claimed that larger crystal grains are produced by the enhanced crystallization of perovskite thin films [42,43].

UV-visible (UV-vis) absorption, steady-state photoluminescence (PL), and time-resolved PL spectroscopies were used to examine the optoelectronic properties of the as-prepared perovskite thin films. The perovskite thin films fabricated with NMP, TEP, and DMF exhibit almost the same absorption range in the visible spectrum, as shown in Figure 4b. This suggests a low density of in-gap defects, which is helpful for absorbing more incoming photons, and as a result, high solar energy conversion is anticipated [44].

The PL spectra in Figure 4c show that all the perovskite thin films have high peak intensities centered at 1.60 eV corresponding to the band-to-band radiative recombination. High PL intensity indicates the presence of a low density of defect states in the bulk or grain boundaries that commonly induce Shockley–Read–Hall (SRH) recombination [45]. This is in agreement with the observed microstructure of the as-prepared perovskite thin films, which show large, smooth, and compact grains with few boundaries. Furthermore, the TRPL spectra in Figure 4d (with logarithmic scale on the *Y*-axis) show that the perovskite thin films had nearly identical bulk lifetimes of charge carriers, indicating a long lifespan for readily collecting photogenerated charge carriers [46,47].

As illustrated in Figure 5, we also determined the optical band gaps of the as-prepared perovskite thin films using Tauc’s equation: ${\left(\alpha hv\right)}^{2}=A\left(hv-E_{g}\right)$. The data showed that the optical band gaps (1.51 eV) of the DMF-based, TEP-based, and NMP-based perovskite thin films are almost identical. As shown in Figure 4b, this low band gap is thought to be advantageous for harvesting more photons in the visible spectrum, which means that a high photocurrent is likely expected [11].

Stack layers of FTO/SnO_2_/MAFAPbI_3_/Spiro-OMeTAD/Au were used in the fabrication of solar cells employing NMP, TEP, and DMF precursor solvents. The current–voltage (J-V) characteristic curves and cross-sectional SEM images of the as-fabricated devices are displayed in Figure 6. The cross-sectional SEM images of the PSCs in Figure 6a–c display perovskite thin film grains that match the surface morphologies depicted in Figure 2. The PCE for the NMP-based PSCs is 20.78% under forward-biased conditions, whereas the PCEs for the TEP-based PSC and DMF-based devices are 20.98% and 20.58%, respectively, according to the J-V characteristic curves (Figure 6d–f). All of the devices have nearly identical photovoltaic parameters, as shown in Table 2. It is speculated that light just passes through the void regions, where there is no perovskite, when the substrate surface is coated with small crystal grains that contain some voids. On the other hand, perovskite thin films with smooth surfaces and fewer grain boundaries absorb more light, leading to better absorption. Smooth surface morphologies with large grains and few grain boundaries contributed to the high performance of the as-prepared devices because they enhance photon harvesting and decrease charge carrier recombination [48].

A higher PCE would result from larger grains in the perovskite films because photogenerated charge carriers would encounter less resistance from bulk defects and grain boundaries. Grain boundaries and/or large pinholes sizes or densities are additionally linked to poor device performance due to a large number of trap-assisted recombination sites of trapping and lower carrier mobilities [43]. Larger crystal grains will improve the overall performance of PSCs since they expose the charge carriers to fewer sources of scattering and trapping. Given that perovskite thin films with larger grains, less grain boundaries, and smoother surface morphologies allow charge carriers to pass through the crystals more efficiently, high photovoltaic performance is subsequently obtained [49].

The charge carrier dynamics within the PSCs under photoexcitation with a reverse bias of 0.5 V and a frequency range of 1 Hz to 1 MHz were examined using electrical impedance spectroscopy (EIS). Nyquist plots with equivalent circuits are displayed in Figure 7a for the purpose of assessing the charge carrier recombination resistances (R_r_) [50]. Since the charge-transporting layers for the as-prepared devices were not changed, the perovskite thin films are the only factor affecting the R_r_. As can be seen in Figure 7a, PSCs based on TEP and DMF have R_r_ values of 54.6 KΩ and 53.6 KΩ, respectively, whereas NMP-based PSCs have an R_r_ value of 53.7 KΩ. This is consistent with the earlier findings and shows that all devices can effectively resist charge carrier recombination [24,47]. In the meantime, the statistical data in Figure 7b and Table 3 for the PSCs demonstrate that all of them have higher photovoltaic parameters and improved repeatability. Table 3 and Figure 7b demonstrate that every photovoltaic metric falls into a narrow range. Based on NMP, the PCEs for four PSCs ranged from 20.05% to 20.78%, with an average of 20.53%. In the same way, the PCEs for the DMF- and TEP-based PSCs have average values of 20.22% and 20.71%, respectively, and are distributed within a narrow range. Another appealing aspect of PSCs is their long-term stability, which goes hand in hand with their high PCEs. To that end, under AM 1.5G irradiation, the stability of both encapsulated and unencapsulated TEP-based PSCs were assessed every 72 h for a total of 720 h. Following each stability test, the measured PSCs were kept at 25 °C and 65% humidity in a glass oven. The stability of the encapsulated PSCs was superior to that of the unsealed devices, as illustrated in Figure 7c. After 720 h of testing, the efficiency of sealed PSCs slightly decreases, going from 20.71% to 20.36%, indicating an absolute value of a 0.35% loss in PCE (1.7% relative loss). The efficiency of the unsealed PSCs dropped sharply from 20.68% to 13.99% under the same testing conditions, indicating poor stability and quick degradation. Fast erosion leads to the degradation of the PSCs because the unsealed devices were exposed to humidity [51]. Additionally, under AM 1.5G, the steady-state J_sc_ and PCE of the TEP-based PSC were monitored at maximum power at a forward bias voltage of 0.90 V at room temperature (Figure 7d). After 236 s of continuous illumination, the TEP-based device showed a stabilized PCE of 20.39%, which is nearly identical to the maximum PCE of 20.98% for the same PSC. These findings demonstrate that the methodology used in this study is feasible for developing more environmentally friendly PSCs as well as high-quality perovskite layers, which lead to PSCs with excellent photovoltaic performance.

Here, we compare the photovoltaic output of PSCs developed through various green methods. The PCE noted here is the highest value for the PSCs created using an environmentally friendly approach to prepare the perovskite layer, as Table 4 illustrates. It is evident that using only one green solvent that dissolves common perovskite precursors while avoiding the need for an antisolvent is challenging. Most investigators need to create complex solvent systems or use non-halide precursors in addition to antisolvents to achieve environmentally friendly perovskite film preparation because lead halide can be challenging to dissolve in common green solvents. In this context, we came up with a novel green strategy to overcome the challenging problem of the toxic host solvents and toxic antisolvents by utilizing only the green solvent of TEP in the two-step approach, which represents a significant advancement towards the achievement of greener PSC fabrication.

## 4. Conclusions

This work demonstrates a novel environmentally friendly method for simultaneously eliminating solvent toxicity from the precursor solvent and antisolvent during the formation of perovskite thin films. The PbI_2_ layer is prepared in the first fabrication step using a non-toxic TEP solvent. To avoid using an antisolvent in the second step, FAMAI-TEP solution is poured on the surface of the PbI_2_ layer. This approach leads to the production of greener planar PSCs with a PCE of 20.98%. On the other hand, PCEs of 20.78% and 20.58% are obtained from PSCs made using NMP- and DMF-based perovskite thin films, respectively. This indicates that the TEP solvent is a more environmentally friendly substitute for the hazardous DMF and NMP precursor solvents. These promising results pave the way for the development of an environmentally friendly and cost-effective process for the production of PSCs without sacrificing device performance.

## Figures and Tables

**Figure 1 nanomaterials-14-00594-f001:**
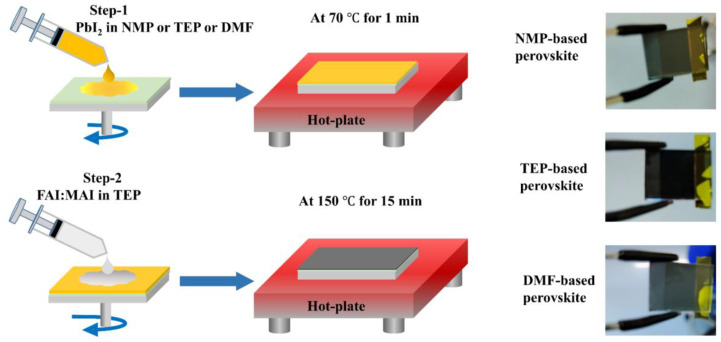
Schematic illustration of the novel greener approach for the fabrication of planar perovskite solar cells.

**Figure 2 nanomaterials-14-00594-f002:**
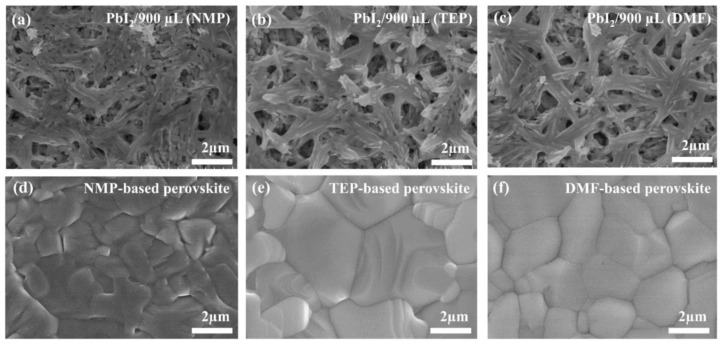
(**a**–**c**) SEM images of the PbI_2_ thin films prepared from the spin-coating of a PbI_2_ precursor solution in NMP, TEP, and DMF, respectively. (**d**–**f**) SEM images of the MAFAPbI_3_ thin films prepared after spin-coating the MAFAI precursor solution in TEP on the surface of the as-prepared PbI_2_ layer.

**Figure 3 nanomaterials-14-00594-f003:**
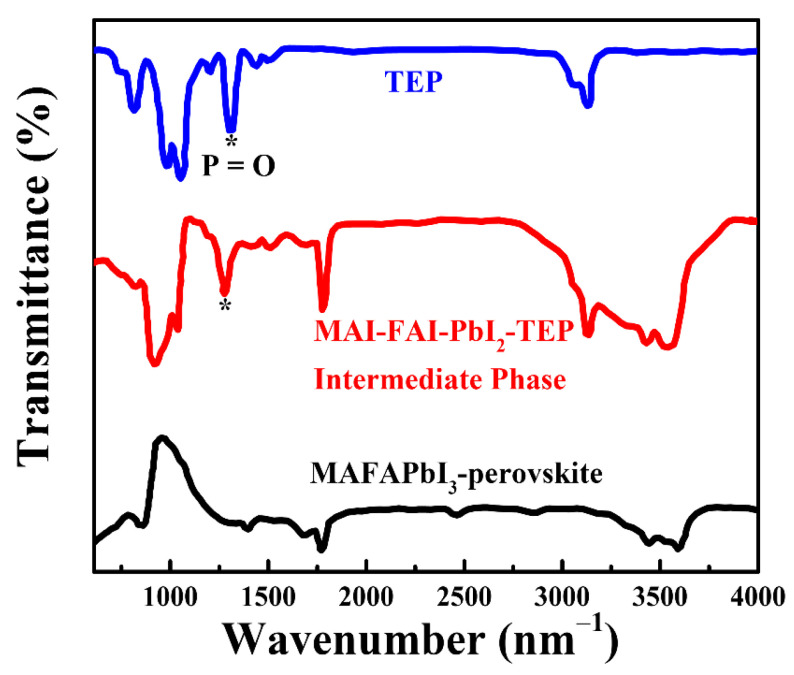
FTIR spectra of the TEP, perovskite’s intermediate phase, and a perovskite thin film prepared on a glass substrate. * Corresponds to the stretching vibration of the P=O bond in TEP.

**Figure 4 nanomaterials-14-00594-f004:**
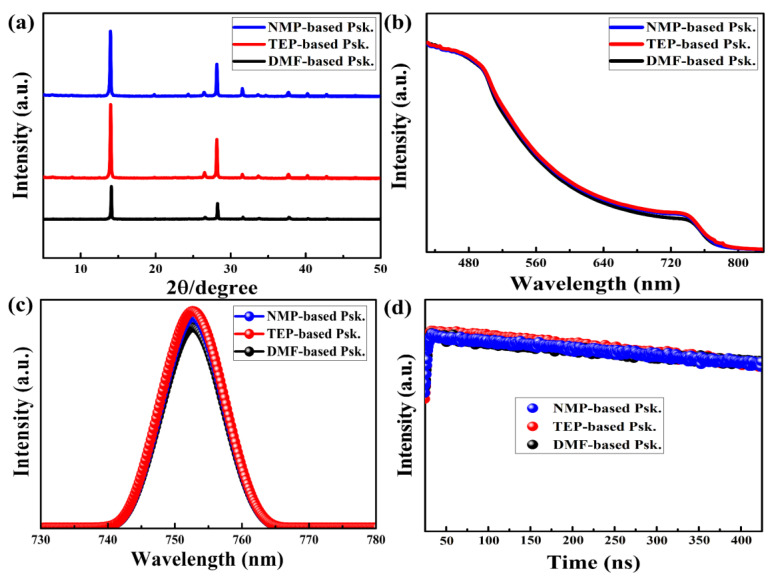
(**a**) XRD data, (**b**) UV-vis absorption spectra, (**c**) PL spectra, and (**d**) TRPL spectra of the as-prepared perovskite thin films on glass substrates.

**Figure 5 nanomaterials-14-00594-f005:**
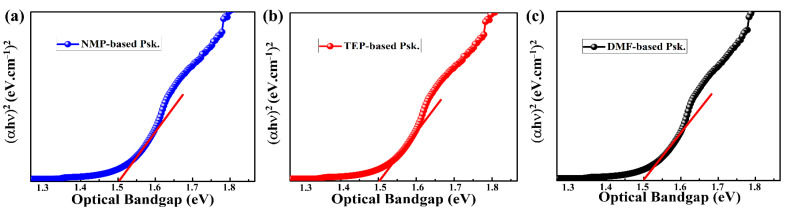
Estimation of optical band gaps of the as-prepared perovskite thin films (**a**) NMP, (**b**) TEP, (**c**) DMF.

**Figure 6 nanomaterials-14-00594-f006:**
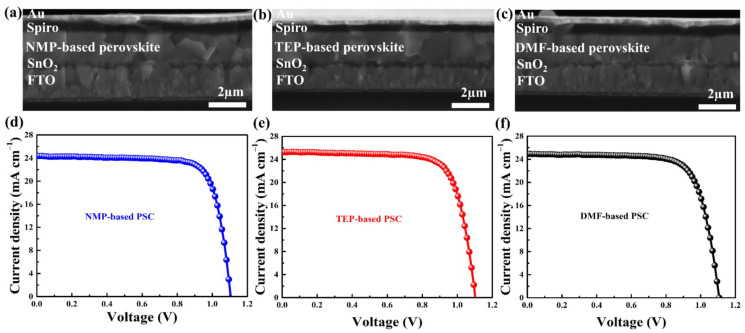
(**a**–**c**) Cross-sectional SEM images and (**d**–**f**) current–voltage (J-V) characteristic curves of the as-fabricated perovskite solar cells.

**Figure 7 nanomaterials-14-00594-f007:**
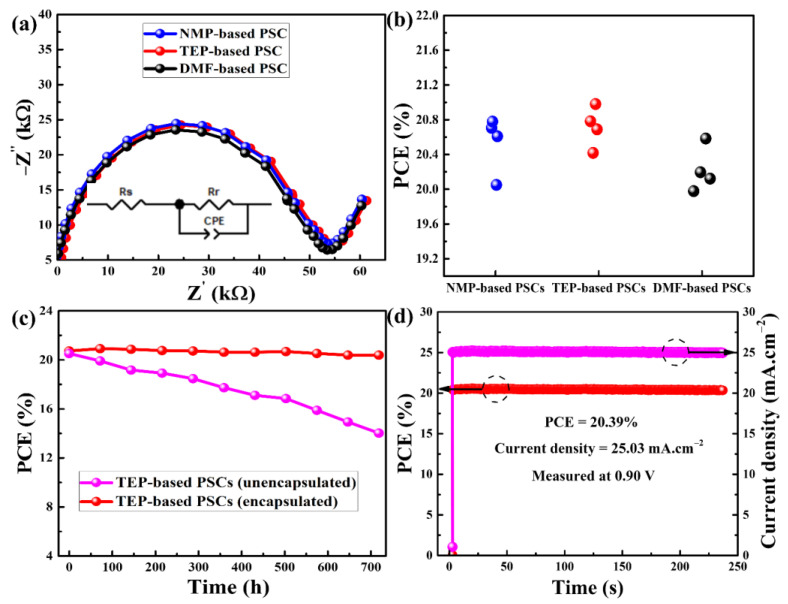
(**a**) Electrical impedance spectroscopy, (**b**) statistical data, and (**c**,**d**) stability analysis of as-fabricated perovskite solar cells.

**Table 1 nanomaterials-14-00594-t001:** Classification of traditional solvents and green solvent used in this work.

Solvent	M.P. (°C)	B.P. (°C)	GHS Symbol
N-Methyl-2-pyrrolidone (NMP)	−24	204	
Dimethylformamide (DMF)	−61	153	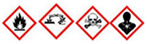
Dimethyl sulfoxide (DMSO)	19	189	
Gamma-butyrolactone (GBL)	−43.53	204	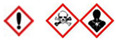
Triethyl phosphate (TEP) (used in this work)	57	215	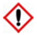

**Table 2 nanomaterials-14-00594-t002:** Photovoltaic parameters of the as-prepared PSCs.

Device	V_oc_ (V)	J_sc_ (mA cm^−2^)	FF (%)	PCE (%)
NMP-based PSC	1.109	24.33	77.00	20.78
TEP-based PSC	1.109	25.25	74.92	20.98
DMF-based PSC	1.109	24.89	74.54	20.58

Abbreviations: V_oc_ (open-circuit voltage), J_sc_ (short-circuit current density), FF (fill factor), and PCE (power conversion efficiency).

**Table 3 nanomaterials-14-00594-t003:** Statistical data for perovskite solar cells fabricated from perovskite precursor solution containing NMP, TEP, and DMF, respectively.

**NMP-Based PSCs**	**V_oc_ (V)**	**J_sc_ (mA cm^−2^)**	**FF (%)**	**PCE (%)**
	1.109	23.69	76.29	20.05
	1.109	24.71	75.19	20.61
	1.109	24.18	77.20	20.70
	1.109	24.33	77.00	20.78
**TEP-based PSCs**	1.122	25.34	71.82	20.42
	1.109	24.36	76.58	20.69
	1.109	25.25	74.22	20.78
	1.109	25.25	74.92	20.98
**DMF-based PSCs**	1.109	23.87	75.44	19.97
	1.109	25.23	71.91	20.12
	1.109	24.04	75.73	20.19
	1.109	24.89	74.54	20.58

**Table 4 nanomaterials-14-00594-t004:** Performance comparison of PSCs produced in environmentally friendly conditions.

Solvent	Deposition Method	Perovskite Precursor	Device Configuration	PCE (%)	Ref.
CAN/MA/IPA	One-step	PbI_2_ + MAI	Planar: FTO/TiO_2_/C_60_/MAPbI_3_/Spiro-OMeTAD/Au	19	[52]
MAAc	One-step	PbI_2_ + MACl	Planar: FTO/SnO_2_/Perovskite/MoO_3_/Spiro-OMeTAD/Au	20.05	[53]
DMI/EA	Blade coating	Pb(Ac)_2_ + MAI	Planar: FTO/SnO_2_/Perovskite/Spiro-OMeTAD/Au	18.26	[54]
GBL/EtOH/AcOH	One-step	Pb(Ac)_2_·3H_2_O + PbCl_2_	Planar: FTO/TiO_2_/Perovskite/Spiro-OMeTAD/Au	15.10	[55]
DMSO/2-MP/1-P	One-step	Pb(Ac)_2_·3H_2_O + PbCl_2_ + MAI	Planar: FTO/TiO_2_/Perovskite/Spiro-OMeTAD/Au	16.50	[56]
MAP/ACN/DMSO/IPA	Two-step	PbI_2_/(FAI + MABr)	Mesoporous: FTO/c-TiO_2_/m-TiO_2_/Perovskite/Spiro-OMeTAD/Au	15.46	[57]
ACN/MA	One-step	(PbI_2_ + MAI)/MACl	Planar: FTO/SnO_2_/Perovskite/Spiro-OMeTAD/Au	18.70	[58]
H_2_O/IPA	Two-step	Pb(NO_3_)_2_/MAI	Mesoporous: FTO/c-TiO_2_/m-TiO_2_/Perovskite/Spiro-OMeTAD/Au	16.70	[59]
H_2_O/IPA	Two-step	Pb(NO_3_)_2_/(FAI + FABr)	Mesoporous: FTO/c-TiO_2_/m-TiO_2_/Perovskite/Spiro-OMeTAD/Au	13	[60]
H_2_O/IPA	Two-step	Pb(NO_3_)_2_/MAI	Mesoporous: FTO/c-TiO_2_/m-TiO_2_/Perovskite/Spiro-OMeTAD/Au	13.7	[61]
H_2_O/IPA	Two-step	Pb(NO_3_)_2_/(MAI + MACl)	Mesoporous: FTO/c-TiO_2_/m-TiO_2_/Perovskite/Spiro-OMeTAD/Au	15.11	[62]
H_2_O/IPA	Two-step	Pb(NO_3_)_2_/MAI	Mesoporous: FTO/c-TiO_2_/m-TiO_2_/Perovskite/Spiro-OMeTAD/Au	12.58	[63]
TEP/DEE	One-step	(PbI_2_ + FAI + MACl)	Planar: FTO/SnO_2_/Perovskite/Spiro-OMeTAD/Au	18.65	[33]
TEP	Two-step	PbI_2_/FAI:MAI	Planar: FTO/SnO_2_/Perovskite/Spiro-OMeTAD/Au	20.98	This work

Abbreviations: ACN: acetonitrile; MA: methylamine; MAAc: methylamine acetate; DMI: 1,3-Dimethyl-2-imidazolidinone; EA: ethyl acetate; GBL: gamma-butyrolactone; EtoH: ethanol; AcOH: acetic acid; DMSO: dimethyl sulfoxide; 2-MP: 2-methylpyrazine; 1-P: 1-pentanol; MAP: propionate; IPA: isopropyl alcohol; TEP: triethyl phosphate; DEE: dibutyl ether.

## Data Availability

All the data presented in the manuscript can be obtained from the corresponding authors upon reasonable request.

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
