# Peer review of "Fabricating Planar Perovskite Solar Cells through a Greener Approach"

_nanomaterials, 2024, doi:10.3390/nano14070594_

Round 1

Reviewer 1 Report

Comments and Suggestions for Authors

This is a very interesting work. The elimination of toxic solvents in fabricating perovskite solar cells is very important to the potential for large-scale manufacturing. The paper is well written and the results support the conclusion. I have just one suggestion:

In Table 2, include the full name of the photovoltaic parameters in the table title or as a footnote just below the table. That is, Voc is open-circuit voltage, Jsc is short-circuit current density,  FF is fill factor, and  PCE is power conversion efficiency.

Comments on the Quality of English Language

The English Language is fine. Only minor editing is needed.

Author Response

Reply to Reviewer 1

Comments and Suggestions for Authors

This is a very interesting work. The elimination of toxic solvents in fabricating perovskite solar cells is very important to the potential for large-scale manufacturing. The paper is well written and the results support the conclusion. I have just one suggestion:

In Table 2, include the full name of the photovoltaic parameters in the table title or as a footnote just below the table. That is, Voc is open-circuit voltage, Jsc is short-circuit current density, FF is fill factor, and PCE is power conversion efficiency.

Comments on the Quality of English Language

The English Language is fine. Only minor editing is needed.

Reply: We sincerely appreciate your opinions and remarks. The revised manuscript includes the abbreviation in accordance with that. Furthermore, the English language is reviewed and adjusted as necessary.

Reviewer 2 Report

Comments and Suggestions for Authors

In this study, the authors have developed high-efficiency planar PSCs utilizing a new eco-friendly technique. Initially, this approach involves replacing harmful precursor solvents with a safer alternative, triethyl phosphate (TEP), through a more sustainable solvent engineering process. Subsequently, they crafted perovskite thin films employing a dual-stage method without requiring any antisolvent. Among the eco-friendlier methods for constructing PSCs, the configuration of the FTO/SnO2/MAFAPbI3/spiro-OMeTAD planar device stood out, achieving the highest power conversion efficiency (PCE) at 20.98%.

1.      The methodology for cleaning the FTO substrates includes a sequence involving ethanol, isopropyl alcohol, deionized water, and a detergent solution. Could the authors provide a rationale for this specific cleaning sequence and explain if they have explored the impact of varying this sequence on the quality and efficiency of the final perovskite films?

2.      Additionally, how do they ensure that no residue from the detergent solution remains on the substrates, which could potentially affect the performance of the solar cells?

3.      Please provide empirical evidence or comparative studies to support the claim that this level of humidity is optimal for the growth of high-quality perovskite crystals and high-performing devices?

4.      How do you the morphological differences correlate with the performance metrics of the PSCs, such as efficiency, stability, and charge carrier mobility?

5.      Since the grain boundaries can significantly affect the efficiency and stability of PSCs, how do you ensure that the XRD results reflect a uniform distribution of crystal sizes across the entire film, rather than localized regions of large grains?

6.      In the study, it is stated that "the charge transporting layers for the as-prepared devices were not changed," implying that the perovskite thin films are the singular elements influencing the recombination resistances (Rr). However, considering the complex nature of PSCs where multiple layers interact, it is unclear to me how the unaltered charge transporting layers do not affect the overall device performance and Rr values.

7.      The following papers related to perovskite should be cited in the introduction to give readers more background information: Nanoscale Advances 3 (23), 6659, 2021; Journal of Materiomics, 7 (5), 940, 2021.

8.      How significant is the variance in efficiency and stability between the encapsulated and unencapsulated TEP-based PSCs in the long term, and could this influence the overall viability and commercial applicability of using TEP as an alternative solvent?

Author Response

Reply to Reviewer 2

All changes are made in red color in the revised manuscript.

Comments and Suggestions for Authors

In this study, the authors have developed high-efficiency planar PSCs utilizing a new eco-friendly technique. Initially, this approach involves replacing harmful precursor solvents with a safer alternative, triethyl phosphate (TEP), through a more sustainable solvent engineering process. Subsequently, they crafted perovskite thin films employing a dual-stage method without requiring any antisolvent. Among the eco-friendlier methods for constructing PSCs, the configuration of the FTO/SnO2/MAFAPbI3/spiro-OMeTAD planar device stood out, achieving the highest power conversion efficiency (PCE) at 20.98%.

  1. The methodology for cleaning the FTO substrates includes a sequence involving ethanol, isopropyl alcohol, deionized water, and a detergent solution. Could the authors provide a rationale for this specific cleaning sequence and explain if they have explored the impact of varying this sequence on the quality and efficiency of the final perovskite films?

Reply: We sincerely value your insightful remarks. This cleaning technique only cleans the surface residue of FTO substrates. Our research's primary goal is to offer an environmentally friendly process to manufacture PSCs. Furthermore, a correction has been made to the revised manuscript regarding the cleaning sequence, which now reads as follows: "The FTO-substrates were cleaned sequentially by sonicating them in detergent solution, deionized water, ethanol and isopropyl alcohol for 20 min each." Lines 115–116 on Page #3 illustrate this.

  1. Additionally, how do they ensure that no residue from the detergent solution remains on the substrates, which could potentially affect the performance of the solar cells?

Reply: Thank you again for bringing this to our attention. In fact, there was an error in the cleaning order, which is fixed on page #3 of the revised manuscript. To further ensure the cleaned surface, the cleaned substrates were placed in a UV ozone cleaning chamber. Furthermore, over the course of the last seven years, we have used this method and have shown that PSCs perform well (references: https://doi.org/10.1002/adma.201707583, https://doi.org/10.1016/j.nanoen.2018.11.004, https://doi.org/10.1016/j.solener.2021.08.015, etc.).

  1. Please provide empirical evidence or comparative studies to support the claim that this level of humidity is optimal for the growth of high-quality perovskite crystals and high-performing devices?

Reply: We thank you for your suggestion very much. As evidenced by publications (https://doi.org/10.1126/science.abp8873, https://doi.org/10.1063/1.4901510, etc.) cited in the manuscript, page#3, lines 125–129, other researchers have already reported on the impact of the humidity level. As a result, the primary focus of our work is solely on how the green approach affects PSC performance.

  1. How do you the morphological differences correlate with the performance metrics of the PSCs, such as efficiency, stability, and charge carrier mobility?

Reply: The morphologies of the resulted perovskite thin-films in our case is compact, pinholes free, and contain large grains with minimum grain boundaries (as discussed and highlighted in red color on page#5, line 182-186). Therefore, larger grain sizes in perovskite films would promote the PCEs of PSCs, since the photogenerated carriers encounter reduced impediments from bulk defects and grain boundaries. Decomposition of perovskite materials usually starts at grain boundaries after long-term illumination, where water and oxygen easily penetrate into the perovskites, resulting in continuous decomposition/degradation because of weaker chemical bonding and ion migration at grain boundaries in low-crystallinity perovskites. In addition to the encapsulation, the stability improvement is originated from the larger grain size with minimum grain boundaries. Furthermore, the few grain boundaries in the resulted perovskite thin-films can effectively alleviate the moisture attacks.  

  1. Since the grain boundaries can significantly affect the efficiency and stability of PSCs, how do you ensure that the XRD results reflect a uniform distribution of crystal sizes across the entire film, rather than localized regions of large grains?

Reply: As the XRD patterns were obtained for the complete perovskite thin-films prepared on glass substrates (the whole substrate were exposed to X-ray), which indicated the pure phase of the perovskite with the (110) and (220) planes at 140 and 280. All of the perovskite thin-films in these samples have excellent crystals with nearly identical intensity peaks, which suggests large grains with few boundaries, based on the XRD analysis of these samples. This is consistent with earlier research that suggested enhanced crystallization of perovskite thin-films produces larger crystal grains (https://doi.org/10.1038/ncomms7142, https://doi.org/10.1038/s41598-018-31184-0). These predictions were further supported by the SEM images of the glass substrates/perovskite thin-films (Fig. 2(d, e, and f)).  

  1. In the study, it is stated that "the charge transporting layers for the as-prepared devices were not changed," implying that the perovskite thin films are the singular elements influencing the recombination resistances (Rr). However, considering the complex nature of PSCs where multiple layers interact, it is unclear to me how the unaltered charge transporting layers do not affect the overall device performance and Rr values.

Reply: We agree with your viewpoint, the EIS response of the device will certainly not be impacted by applying the same interface materials (charge transport materials) in the same quantity and fabrication method. However, as shown by the data in Fig. 7(a), the Rr is slightly altered when perovskite layers are fabricated using different solvents for the same device structure and interface materials.

  1. The following papers related to perovskite should be cited in the introduction to give readers more background information: Nanoscale Advances 3 (23), 6659, 2021; Journal of Materiomics, 7 (5), 940, 2021.

Reply: Thank you for your suggestion. We cited the mentioned articles accordingly in the revised manuscript (page#1, line#37).

  1. How significant is the variance in efficiency and stability between the encapsulated and unencapsulated TEP-based PSCs in the long term, and could this influence the overall viability and commercial applicability of using TEP as an alternative solvent?

Reply: We value your opinions regarding the TEP as a substitute solvent. Our research is primarily focused on using TEP as a green solvent while also using a two-step approach to avoid using an antisolvent. In addition to the study, the devices' stability was examined to determine how they functioned at 25°C and 65% relative humidity. Based on our analysis of the stability of the devices, we anticipate that once the encapsulation techniques used in commercial devices for other relevant technologies are adopted, encapsulated devices will last a long time and be commercially viable.

Reviewer 3 Report

Comments and Suggestions for Authors

This work describes the formation, characterization, and functionalization of perovskite films fabricated by employing greener solvent engineering. The morphological, structural, and optical properties of the films were detailed and presented. Overall, the results are good and relevant in the context of perovskite solar cells. Nevertheless, some parts of the paper need improvement. I suggest the following revisions to enhance the quality of the manuscript:

1. The authors claim the use of environmentally friendly solvents to decrease the toxicity of materials used in the production of perovskites. However, it is important to highlight that hybrid organic-inorganic perovskites often contain lead, which poses a significant environmental and health concern. Comments addressing this issue should be added to highlight the challenge of mitigating lead toxicity in the context of upgrading this technology.

2. If perovskite formation by solution processing has the drawback of using hazardous solvents and antisolvents, the authors are encouraged to consider vapor processing as an alternative method. A more comprehensive introduction section should describe the pros and cons of solution processing versus vapor processing. High-purity films have been demonstrated to be formed by vapor deposition. References supporting the formation of high-quality perovskite films by vapor deposition should also be included (https://doi.org/10.1016/j.tsf.2018.08.026; https://www.nature.com/articles/srep29910). Additionally, to fabricate planar solar cells, the advantages and disadvantages of solution processing versus physical vapor deposition (PVD) methods should be discussed.

3. The authors should explain the rationale behind presenting detailed values of masses used for each perovskite precursor, such as why 691.6 mg of PbI2 was used. Justify the chosen values to enhance reproducibility and transparency in the experimental procedure. References supporting the chosen values, if available, should be cited.

4. The authors must clarify the statement regarding the necessity of humidity to improve the performance of perovskite films. While humidity can influence perovskite film properties, it is essential to acknowledge that excessive humidity can lead to film degradation. Discuss the balance between humidity effects on perovskite performance and the risk of decomposition due to prolonged exposure.

5. Please address whether there are differences in the color of the samples based on the precursors used and report the band gap of the films to provide insights into their optical properties.

6. The authors are advised to provide uncertainties for the photovoltaic parameters of the as-prepared perovskite solar cells and specify the number of samples used for the reported values to facilitate a better understanding of the experimental variability.

7. The authors are requested to improve the clarity and specificity of the conclusion section by providing a more detailed summary of the key findings and their implications. Avoid vague language and ensure that the conclusions are supported by the results presented in the manuscript.

Comments on the Quality of English Language

Minor editing of English language required.

Author Response

Reply to Reviewer 3

All changes are made in red color in the revised manuscript.

Comments and Suggestions for Authors

This work describes the formation, characterization, and functionalization of perovskite films fabricated by employing greener solvent engineering. The morphological, structural, and optical properties of the films were detailed and presented. Overall, the results are good and relevant in the context of perovskite solar cells. Nevertheless, some parts of the paper need improvement. I suggest the following revisions to enhance the quality of the manuscript:

  1. The authors claim the use of environmentally friendly solvents to decrease the toxicity of materials used in the production of perovskites. However, it is important to highlight that hybrid organic-inorganic perovskites often contain lead, which poses a significant environmental and health concern. Comments addressing this issue should be added to highlight the challenge of mitigating lead toxicity in the context of upgrading this technology.

Reply: We sincerely appreciate your recommendation. The introduction section, page #1, and lines 43–49 of the revised manuscript now include the remarks regarding lead toxicity.

  1. If perovskite formation by solution processing has the drawback of using hazardous solvents and antisolvents, the authors are encouraged to consider vapor processing as an alternative method. A more comprehensive introduction section should describe the pros and cons of solution processing versus vapor processing. High-purity films have been demonstrated to be formed by vapor deposition. References supporting the formation of high-quality perovskite films by vapor deposition should also be included (https://doi.org/10.1016/j.tsf.2018.08.026; https://www.nature.com/articles/srep29910). Additionally, to fabricate planar solar cells, the advantages and disadvantages of solution processing versus physical vapor deposition (PVD) methods should be discussed.

Reply: We appreciate your recommendation. The revised manuscript includes additional comments on vapor deposition and cites the mentioned articles (see page #2, lines 80–85).

  1. The authors should explain the rationale behind presenting detailed values of masses used for each perovskite precursor, such as why 691.6 mg of PbI2 was used. Justify the chosen values to enhance reproducibility and transparency in the experimental procedure. References supporting the chosen values, if available, should be cited.

Reply: We appreciate your insightful notice. PbI2 is used in the amount reported in our previously published articles (https://doi.org/10.1002/adma.201707583, https://doi.org/10.1016/j.apsusc.2020.148583, https://doi.org/10.1021/acsaem.0c00563 etc.) and in the article (https://doi.org/10.1126/science.abp8873) that describes the two-step process in which PbI2 is used (cited page#3, line # 106).

  1. The authors must clarify the statement regarding the necessity of humidity to improve the performance of perovskite films. While humidity can influence perovskite film properties, it is essential to acknowledge that excessive humidity can lead to film degradation. Discuss the balance between humidity effects on perovskite performance and the risk of decomposition due to prolonged exposure.

Reply: We appreciate your suggestion very much. Previous studies (https://doi.org/10.1126/science.abp8873, https://doi.org/10.1063/1.4901510, etc. cited in the manuscript, page#3, line 123-129) have already reported on the effect of the humidity level. As a result, we limit our attention to the primary goal of our work, which is the effects of the green approach on PSC performance.

  1. Please address whether there are differences in the color of the samples based on the precursors used and report the band gap of the films to provide insights into their optical properties.

Reply:  We have not noticed any colour changes in the as-prepared perovskite thin-films. In addition, the optical band gaps of the perovskite thin-films were calculated by using Tauc’s equation in the revised manuscript, as shown in Fig. 5 page#8.

  1. The authors are advised to provide uncertainties for the photovoltaic parameters of the as-prepared perovskite solar cells and specify the number of samples used for the reported values to facilitate a better understanding of the experimental variability.

Reply: We thank your suggestion so much. The reproducible nature of the as-prepared solar cells are demonstrated by the statistical data in Table 3 and Fig. 7(b). Four devices using NMP-based perovskite thin films, four using TEP-based perovskite thin films, and four using DMF-based perovskite thin films had their data calculated (Page#10).

  1. The authors are requested to improve the clarity and specificity of the conclusion section by providing a more detailed summary of the key findings and their implications. Avoid vague language and ensure that the conclusions are supported by the results presented in the manuscript.

Reply: Thank you for your valuable suggestion. The conclusion section is modified accordingly in the revised manuscript.

Comments on the Quality of English Language

Minor editing of English language required.

Reply: The manuscript is scrutinized and edited as needed to improve the English language wherever necessary.

Round 2

Reviewer 2 Report

Comments and Suggestions for Authors

Paper can be published as it is.

Reviewer 3 Report

Comments and Suggestions for Authors

The authors have addressed the Reviewer comments satisfactorily. This manuscript is of interest to the journal, and I recommend its acceptance.

Comments on the Quality of English Language

Minor editing of English language required.